# Automata Learning from Recurrent Networks: A Critical Synthesis for Verification, Testing, and Interpretability

**Jaouhar Slimi**                                     *jaouhar.slimi@cea.fr*
*Université Paris-Saclay, CEA, List, France*

**Augustin Lemesle**                                  *augustin.lemesle@cea.fr*
*Université Paris-Saclay, CEA, List, France*

**Tristan Le Gall**                                   *tristan.le-gall@cea.fr*
*Université Paris-Saclay, CEA, List, France*

**Zakaria Chihani**                                   *zakaria.chihani@cea.fr*
*Université Paris-Saclay, CEA, List, France*

**Reviewed on OpenReview:** *https://openreview.net/forum?id=R52ETbUBVo*

## Abstract

Recurrent Neural Networks (RNNs) have demonstrated their effectiveness in modeling sequential data and are a key building block of modern deep learning architectures. In this review paper, we study recurrent networks through the lens of automata theory. Given an RNN, automata learning seeks to model its behavior with an automaton, which enables better interpretability and eases our understanding of its working mechanisms. We begin by examining the theoretical foundations of this approach, demonstrating how it can be applied to learn automata from various types of recurrent architectures, including the Elman Recurrent Network (ERN), Long Short-Term Memory (LSTM), and Gated Recurrent Unit (GRU). Next, we review the applications of this approach in formal verification, model-based testing, and the interpretability of these deep learning models. We finish with a discussion on the advantages and critical problems of this method, while outlining key goals for future research, such as defining standard benchmarks and identifying limitations that need to be addressed to advance this field further.

## 1 Introduction

The widespread success of deep learning algorithms and their adoption in various sectors such as healthcare, transportation, and energy are well-established. However, such algorithms are often trained on large datasets, with the goal of optimizing some objective function, and are shallowly evaluated by measuring how close their predictions are to the ground truth. Although deep learning models are capable of performing many tasks, such capabilities are not fully understood by researchers. For instance, numerous failures have been reported, showcasing their fragility in real-world settings (McGregor, 2020). The lack of theoretical understanding of how these models operate internally makes their deployment in such safety-critical domains precarious. To this end, many efforts have been dedicated to developing theories, algorithms, and tools to enhance the safety and interpretability of deep learning models.

Recurrent neural networks are deep learning models that are effective in sequential learning tasks due to their ability to efficiently capture temporal dependencies in data. With the added benefit of their compactness and fast inference, they received some attention, most notably in applications involving time series analysis, sequential planning, and natural language processing, to name a few. Although successful in many domains, the internal behavior of these neural networks is not fully understood. Finding a human-interpretable

explanation of how they operate is a challenging problem due to their complex architecture and high dimensionality. In addition, these models can sometimes lack robustness (Papernot et al., 2016), or suffer from memorization and privacy concerns (Yang et al., 2021) in some scenarios. Given these challenges, ensuring that recurrent neural networks are trained to perform their tasks efficiently and reliably is a stepping stone toward safe artificial intelligence. Automata come as natural surrogate models to describe RNN behavior, given their stateful nature, *i.e.,* RNNs process data sequentially by updating a hidden state.

**Brief historical perspective**  Historically, the relationship between automata and recurrent neural networks (artificial neural networks, broadly speaking) is as old as the fields themselves (Arbib, 1987; Perrin, 1995). Both disciplines were founded on the seminal work of McCulloch & Pitts (1943), which inspired decades of separate research in these two fields, up until their convergence in the definition and analysis of RNN architectures (Giles et al., 1989). Automata have proven instrumental in enhancing recurrent networks expressivity of formal languages (Giles et al., 1992; Andrews et al., 1995). With the evolution of RNNs, their capabilities have expanded beyond just modeling formal languages. Modern RNN architectures are capable of solving hard computational challenges, including processing, classifying, and generating voice, text, and video. Consequently, the application of automata theory to RNNs has evolved from being a mere knowledge acquisition tool to becoming a methodology for understanding how these models work. This approach is formally known as *automata learning.*

**Automata learning**  Automata learning (also called model learning (Vaandrager, 2017) or grammatical inference (de La Higuera, 2010)) is a subfield of machine learning and theoretical computer science. Given a system under learning, the goal is to infer a formal model describing it; typically, a deterministic finite automaton. This technique has found numerous applications, most notably in software and hardware verification and validation. The system under learning can be completely a black box; in which case, the automaton is induced from a set of observations, achieved by querying the system. Otherwise, in a white box setting, the observations are constructed by inspecting its internal structure and creating a set of execution traces from which we induce the automaton. Regardless of the learning approach, the goal is to yield a minimal automaton that generalizes the observed behaviors.

The literature on automata learning from recurrent neural networks is rich in methods developed specifically to manage their characteristics. The survey by Jacobsson (2005) extensively studies rule extraction from RNNs and how the field has evolved since the early 1990s. We refer to such methods as passive automata learning in this work. A later review was proposed (Bollig et al., 2022), in which the authors focus on active automata learning methods. Also, more recently, there has been a revived interest in applying automata learning as an approach to simplify recurrent neural network behaviors, including, but not limited to, simple RNNs and gated RNNs such as LSTMs and GRUs. Moreover, the extracted automata have been proven useful to analyze RNNs, whether to explain, test, or verify them. While alternative methodologies for RNN verification (Jacoby et al., 2020; Ryou et al., 2020) and interpretability (Strobelt et al., 2016; Ismail et al., 2021) have been proposed, these approaches frequently face scalability and precision trade-offs, inherent in the high dimensionality and recurrence of RNNs. Consequently, approximating the RNN with a surrogate model offers a viable path to circumvent these complexity barriers. By leveraging automata — which are inherently interpretable and mathematically rigorous — automata learning bridges this gap, providing a mechanism to simultaneously interpret RNN behavior and verify its properties. This paper reviews the research devoted to applying automata learning to recurrent models. Furthermore, this field is growing rapidly. Recent studies have even expanded these techniques to other neural network architectures, such as Peng et al. (2026); Bhattamishra et al. (2026); Song et al. (2024); Zhang et al. (2024); Adriaensen & Maene (2025); Zhang et al. (2025) for transformer-based models (Vaswani et al., 2017), and Xu et al. (2021) for convolutional neural networks (Schmidhuber, 2014).

**Objectives and outline**  This survey makes three contributions to the field. First, it structurally defines the types of learning algorithms (passive, active, and hybrid), and systematically covers the substantial body of work that has emerged since the last two surveys by Bollig et al. (2022) and Jacobsson (2005), while highlighting recent extensions to other architectures such as transformers and convolutional networks. Second, it provides a unified treatment of automata learning from RNNs through the lens of safety, encompassing

verification, testing, and interpretability—whereas prior surveys treat automata learning in isolation from its applications, and only focus on a single learning style. Third, it identifies concrete limitations and open problems in this field, offering a roadmap for future research at the intersection of automata learning and AI safety. In this regard, the structure of this paper is as follows: we first introduce the necessary background in neural networks and automata theory, which we will use throughout the paper in Section 2. Then, we review the theoretical foundations of automata learning from recurrent neural networks in Section 3. This will lay the groundwork for the next Section 4, in which we delve into the applications of this method in verification, testing, and interpretability. Next, we discuss the limitations of automata learning and key challenges to be tackled in future research in Section 5.

Through this comprehensive synthesis, we aim to contribute to the ongoing dialogue on making AI systems, particularly recurrent networks, more reliable, interpretable, and trustworthy.

## 2 Background and Notation

This section introduces the necessary background. For convenience, a complete summary of the notation used throughout this paper is provided in Appendix A.

### 2.1 Recurrent Neural Networks

**Definition 1 (Recurrent Neural Network)** *A recurrent neural network* $\mathcal{R} : \mathbb{R}^{d_{in}} \to \mathbb{R}^{d_{out}}$ *is a composition of two functions: a state transition function* $F_\theta$, *and a readout function* $G_\phi$.

$$h_t = F_\theta(x_t, h_{t-1}) \tag{1}$$
$$y_t = G_\phi(h_t) \tag{2}$$

*$F$ is a recurrent function parameterized by its trainable weight matrix $\theta$, this function receives an input sequence* $\mathcal{X} = (x_1, x_2 \ldots x_T)$ *where* $x_t \in \mathbb{R}^{d_{in}}$ *and produces a sequence of hidden states* $\mathcal{H} = (h_1, h_2 \ldots h_T)$, *where* $h_t \in \mathbb{R}^{d_h}$.[1] *The readout function $G$, parameterized by its trainable weight matrix $\phi$, takes the hidden state value at a time step t and computes a prediction* $y_t \in \mathbb{R}^{d_{out}}$.

The above serves as a generic definition of a recurrent neural network. While we consider the case of a many-to-many mapper, which is the standard case, it can be naturally generalized to other RNN configurations, such as processing a sequence to produce a single output (many-to-one) or generating a sequence from a single input (one-to-many), depending on the requirements of the task. Several recurrent architectures have been introduced and developed to address specific challenges[2], most notably simple architectures, such as Elman RNN (Elman, 1990), and gated architectures such as LSTM (Hochreiter & Schmidhuber, 1997) and GRU (Cho et al., 2014).

### 2.2 Automata Theory

**Definition 2 (Alphabet, Words and Languages)** *Given a non-empty, finite set $\Sigma$, called the* alphabet, *we consider finite sequences of letters* $\sigma_i \in \Sigma$, *called* words $w = \sigma_1 \sigma_2 \ldots \sigma_n$ *with* $n \geq 0$. *The length of a word is denoted by* $|w| = n$. *The set of all finite sequences of letters is denoted by* $\Sigma^*$, *and a* language $\mathcal{L} \subseteq \Sigma^*$ *is any set of words.*

**Definition 3 (Finite Automaton)** *A finite automaton $\mathcal{A}$ is a quintuple* $\langle \Sigma, Q, Q_I, Q_F, \delta \rangle$ *such that $\Sigma$ is the alphabet, composed of input symbols $\sigma$. $Q$, $Q_I \subseteq Q$, and $Q_F \subseteq Q$ are, respectively, finite sets of states, initial states, and final (or accepting) states. Finally, $\delta : Q \times \Sigma \to Q$ is a state transition function.*

**Definition 4 (Path)** *Given an automaton $\mathcal{A}$ and a word $w$, the path $\pi_w$ of $w$ in $\mathcal{A}$ is a finite sequence of consecutive transitions, defined as:* $q_0 \xrightarrow{\sigma_1} q_1 \ldots \xrightarrow{\sigma_n} q_n$ *or* $q_0 \xrightarrow{\sigma_1 \ldots \sigma_n} q_n$.

---

[1]RNNs are typically initialized with a hidden state value $h_0 \in \mathbb{R}^{d_h}$, which may be fixed at zero or, often, treated as a learnable parameter.

[2]Scardapane (2025) is an excellent reference to understand recurrent neural networks and their landscape in more depth.

Finite automata (FA), along with other kinds of automata, are commonly used mathematical models for language representations, allowing us to understand how sequences of letters can be recognized, generated, or transformed. For instance, we give an example of a finite automaton in Figure 1 capable of recognizing the following language $\mathcal{L} = \{w \in \Sigma^* | w \in (ab)^*\}$.

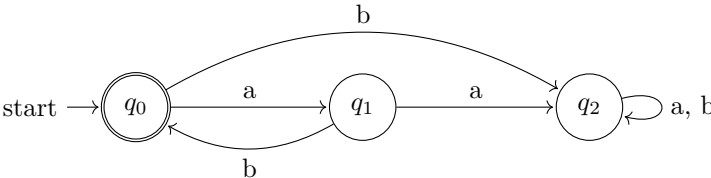

Figure 1: Example of an FA $\mathcal{A} = \langle \Sigma = \{a, b\}, Q = \{q_0, q_1, q_2\}, Q_I = \{q_0\}, Q_F = \{q_0\}, \delta \rangle$

The papers analyzed in this study employ various types of automata, selected based on their specific functional capabilities. While some applications require only a structural state behavior, others necessitate a quantified representation of the system. To ensure this work is self-contained, we briefly outline these automata and their hierarchical relationship to the FA.

We begin with the Labeled Transition System (LTS). Structurally, an LTS resembles an FA where the number of states can be infinite, but it lacks 'accepting' (or final) states; it is typically used to model the ongoing behavior of a system rather than a terminating computation. To introduce quantitative aspects, we use Weighted Finite Automata (WFA), which extend the FA by assigning numerical weights to states and transitions. The Probabilistic Finite Automaton (PFA) is a specific restriction of the WFA where only the transitions are weighted, and the weights are strictly stochastic — values must lie between 0 and 1, and the outgoing transitions from any given state must sum to 1. A further restriction is the Discrete Time Markov Chain (DTMC). A DTMC can be viewed as a PFA that eliminates the input alphabet entirely, meaning transitions represent inherent probabilities of evolution without external triggers. Finally, the Markov Decision Process (MDP) extends the DTMC by reintroducing inputs in the form of 'actions'. This combines the stochastic nature of the DTMC with non-deterministic decision-making.[3]

## 3 Automata Learning from Recurrent Neural Networks

Automata learning from recurrent neural networks involves constructing formal models that represent the behavior of RNNs. We categorize the learning approaches into three types. *Passive learning* often requires full access to the RNN to collect its execution traces on a given sample (typically sequential data), then uses it to yield an automaton. *Active learning* does not necessarily require access to the RNN internals; however, it interacts with the neural network by receiving its feedback (decisions) on a set of inputs. *Hybrid learning* combines active and passive learning in a Counterexample-Guided Abstraction Refinement approach. In this section, we discuss the necessary details of these three learning styles. In addition, we highlight recent works that tackled some of their challenges.

### 3.1 Passive Learning

Passive automata learning infers automata from a set of samples. In the context of recurrent neural networks, passive learning is largely based on the clustering hypothesis (Jardine & van Rijsbergen, 1971)—semantically similar information tends to cluster together. Muskardin et al. (2024) and Michalenko et al. (2019) confirmed this hypothesis, for the case of regular languages, by examining the relationship between semantic ground truth and RNN hidden states. Moreover, Oliva & Lago-Fernández (2021) and Wang et al. (2022) suggest that passive learning benefits significantly from regularization techniques, which force the RNN to behave more like an automaton by making its internal states more stable[4].

---

[3]Sokolova & de Vink (2004); Pouly (2019) provide additional background on quantitative and probabilistic automata.

[4]Stability can be seen as a property of the RNN such that adding noise to some input $x_t$ does not drastically change the state transition dynamics.

The learning procedure is achieved in 4 steps:

1. **Execution traces collection:** we start by feeding input samples $\mathcal{X}$ to the RNN and collecting its hidden states $\mathcal{H}$ as well as its predictions $\mathcal{Y}$, by which we construct the execution traces $(\mathcal{X}, \mathcal{H}, \mathcal{Y})$.

2. **Hidden states clustering:** collected hidden states $\mathcal{H}$ are clustered into a finite set $C = \{c_1, c_2 \ldots c_k\}$, ensuring that $\bigcup_{i=1}^{k} c_i = \mathcal{H}$ and $\forall i \neq j : c_i \cap c_j = \emptyset$. Each cluster $c_i$ forms an abstract state of the automaton, and can be considered as an abstraction of the RNN concrete states $h_t$. Similarly, we designate the transitions of the RNN as *concrete transitions* and those of the automaton as *abstract transitions*. The cluster class is determined by the topmost decision class led by its constituent hidden states.

3. **Pattern extraction:** after obtaining the abstract states, we can adapt the execution traces by assigning each hidden state $h_t$ to its corresponding cluster.

4. **Automaton construction:** the extracted patterns are assembled into an automaton $\mathcal{A}$ such that its abstract states are the obtained clusters $C$, and the abstract transition function $\delta$ is obtained by aggregating the collected concrete transitions. More precisely, a transition exists from abstract state $c_i$ to $c_j$ on input symbol $\sigma$ if the RNN was observed moving from a concrete state in $c_i$ to a concrete state in $c_j$ upon processing $\sigma$. Formally, the set of input symbols enabling a transition between clusters $c_i$ and $c_j$ is defined as: $\Sigma_{i \to j} = \{x_t \in \mathcal{X} | \exists h_t \in c_i, h_{t+1} \in c_j : h_t \to x_t \to h_{t+1}\}$.

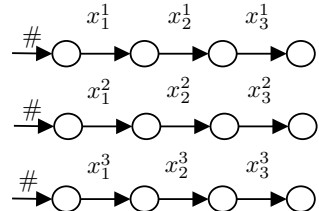

Step 1: Execution traces collection

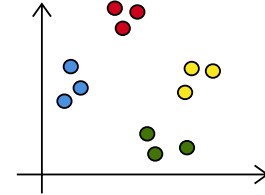

Step 2: Hidden states clustering

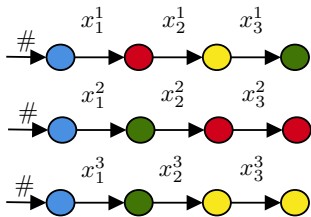

Step 3: Pattern extraction

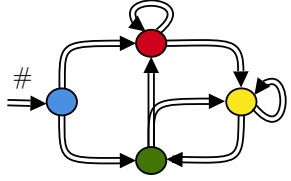

Step 4: Automaton construction

Figure 2: Passive automaton learning from RNN

Consequently, the quality of the learned automata, that is, their size and fidelity to the original RNN behavior, is heavily dependent on two factors: the sample $D$, and the clustering technique. The sample $D$ plays an important role since it is critical to cover as much of the RNN semantics as possible, hence it should be statistically significant. While clustering is critical for automaton precision, it can fail to find the optimal clusters, whether by creating redundant clusters or mixing concrete states that are semantically different in one cluster. To address these limitations, later works have proposed a few methods ranging from algorithmic refinement strategies to novel clustering heuristics.

Zhang et al. (2021) proposes a state abstraction method that determines the optimal number of clusters through the analysis of Decision Confidence Patterns. By discretizing the RNN output probabilities into intervals, the method captures the top prediction classes along with their confidence levels. This allows for

an estimation of the cluster number based on the diversity of observed patterns in the RNN decision traces. Additionally, the approach incorporates context-aware state composition to enhance inference accuracy and utilizes synonym transition methods to handle unseen data.

AdaAX's (Hong et al., 2022) key contribution is the use of adaptive states. After collecting the execution traces and applying standard clustering methods (such as k-means) to the hidden states, the process begins by identifying 'core sets' within the clusters; small groups of hidden states that share both proximity and identical transition behaviors. These core sets are then merged during a consolidation phase, where the merge only happens if the resulting drop in fidelity is less than a user-defined threshold, allowing a trade-off between automaton fidelity and size. Experimental results on both synthetic and real-world datasets, such as Yelp reviews (Zhang et al., 2015), demonstrate that AdaAX achieves higher fidelity while remaining significantly smaller in size compared to standard passive learning.

Merrill & Tsilivis (2022) took a related but distinct approach. Their work extends Gold's algorithm (Gold, 1967) to learn FA from RNNs trained on Tomita regular languages (Tomita, 1982). They first create a prefix tree automaton using the execution traces collected from the RNN, such that the prefix tree automaton states store the RNN hidden states and its output decisions (accept or reject), and the transitions are labeled by the inputs. Second, they launch a state merging algorithm to yield an FA from the prefix tree automaton, by only merging states when the distance between their associated hidden states does not exceed a certain threshold. Their experiments have demonstrated the effectiveness of this approach compared to passive learning with k-means clustering (Wang et al., 2017).

## 3.2 Active Learning

In contrast with passive learning algorithms, active learning does not require a set of execution traces, instead it relies on querying the system under learning, hence such an approach can extract automata from black box RNNs with no access to their internal structure. Active learning algorithms are often variants of the $L^*$ algorithm (Angluin, 1987). $L^*$ is a constructive procedure based on the Myhill-Nerode theorem (Sipser, 2013), which states that a language is regular if and only if the equivalence relation $\sim_{\mathcal{L}}$ has a finite number of equivalence classes. The idea behind $L^*$ is to find these classes, thus finding the automaton describing that language.

$L^*$ operates in a minimally adequate teacher framework, where we have two entities: *the learner* and *the teacher*, as shown in Figure 3. The learner seeks to find the language the teacher has (the RNN in our case), by constructing a hypothesis automaton $\mathcal{A}_{\mathcal{H}}$ from an observation table $O_T$, then asking the teacher if the hypothesis is correct. The learner can ask two types of queries:

**Membership queries (MQs)** Does a word (or a sequence) belong to the teacher's language $\mathcal{L}$?

**Equivalence queries (EQs)** Is the learned hypothesis equivalent to the one the teacher has?

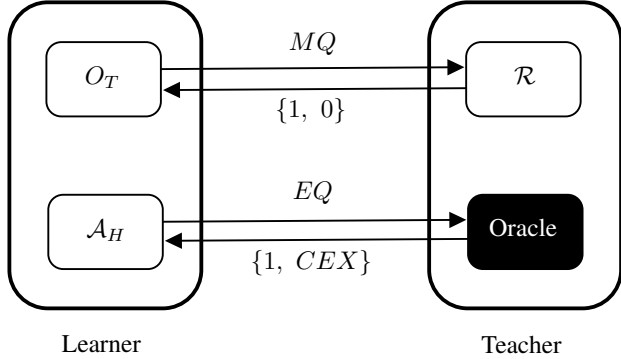

Figure 3: Active automaton learning from RNN

We illustrate the $L^*$ algorithm with the following example, showing how the language $\mathcal{L} = \{a^+\}$ is learned as depicted in Figure 4. For the sake of simplicity, we assume that $\Sigma = \{a, b\}$.

1. We start with an observation table $O_T$. The rows of this table are indexed by $S$, a prefix-closed set of strings (colored in light grey in our figure), and their one-letter extensions $S \cdot \Sigma$ (colored in light blue in the provided example). The columns of $O_T$ are indexed by $E$, which is a suffix-closed set of distinguishing experiments (represented in white). The entry[5] at row $s$ and column $e$ stores the answer to the membership query "Is $s \cdot e \in \mathcal{L}(\mathcal{R})$?"—that is, $O_T(s, e) = 1$ if the RNN accepts the concatenation $s \cdot e$, and 0 otherwise.

2. The learner ensures that $O_T$ satisfies two conditions: *closed* and *consistent*.
   - ◇ A table is closed when $\forall s' \in S \cdot \Sigma \; \exists s \in S \; row(s') = row(s)$; if not, it means it was violated by some word $w \in S \cdot \Sigma$, so we add $w$ to $S$ and its one-letter extensions to $S \cdot \Sigma$ and restart the process with the new sets. In our example, the initial table is not closed since $row(a) = 1$ while $S$ contains only $row(\epsilon) = 0$, so we reconstruct the table by adding $a$ to $S$.
   - ◇ A table is consistent if $\forall s_1, s_2 \in S \; row(s_1) = row(s_2) \implies \forall \sigma \in \Sigma \; row(s_1 \cdot \sigma) = row(s_2 \cdot \sigma)$; if not, we extend the set $E$ by adding the prefixes of the word that led to this inconsistency. In our second example, the table is not consistent since $row(\epsilon) = row(b)$, yet $row(a) \neq row(ba)$, so we extend $E$ by adding $a$ and obtain a consistent observation table.

3. The consistency condition guarantees that the learned FA is deterministic, while the closedness ensures that automaton states are either accepting or rejecting. Thus, we can build a deterministic FA from $O_T$, with as many states as there are rows indexed by $S$, then minimize it to obtain the hypothesis automaton $\mathcal{A}_{\mathcal{H}}$.

4. The learner then sends an equivalence query to the teacher, to know if $\mathcal{L}(\mathcal{A}_{\mathcal{H}})$ is equivalent to $\mathcal{L}(\mathcal{R})$.
   - ◇ If the answer is "Yes", then the learner has identified the language, and the algorithm terminates.
   - ◇ Otherwise, the teacher replies by a counterexample $CEX$ in the symmetric difference of $\mathcal{L}(\mathcal{A}_{\mathcal{H}})$ and $\mathcal{L}(\mathcal{R})$, *i.e.*, $(\mathcal{L}(\mathcal{A}_{\mathcal{H}}) \setminus \mathcal{L}(\mathcal{R})) \cup (\mathcal{L}(\mathcal{R}) \setminus \mathcal{L}(\mathcal{A}_{\mathcal{H}}))$, then $CEX$ is added (as well as all its prefixes) to $S$ and the process starts over with the new sets $S$ and $S \cdot \Sigma$. In our example, after the first hypothesis, the counterexample $ba$ was found, so $b$ and $ba$ were added to $S$.

Within the active learning framework, membership queries for RNNs are manageable, since the network acts as a black-box oracle that classifies input sequences through a standard forward pass. In contrast, equivalence queries pose a significant challenge, as exhaustive equivalence checking is often undecidable or computationally intractable over infinite input spaces. To address this, several approximation techniques have been developed. These include Probably Approximately Correct (PAC) learning methods, which provide probabilistic guarantees of equivalence given sufficiently large test samples, and abstraction refinement techniques[6]. For a more detailed analysis of these strategies, the work by Bollig et al. (2022) reviews state-of-the-art methodologies and theoretical foundations of active learning for RNNs.

Here, we focus on a more recent solution that employs formal conformance testing (Muskardin et al., 2022), in which they combine the W-method (Chow, 1978) with a random walk heuristic to achieve model-guided coverage. The W-method is used to test equivalence based on exhaustive search over all the possible words, assuming an upper bound on the number of states. 'W' refers to the characterization set (or distinguishing set), which is a set of input sequences capable of distinguishing any pair of distinct states in the automaton. Formally, for any two states $q$ and $q'$, there exists a word $w \in W$ such that the output behavior satisfies $\mathcal{A}_{\mathcal{H}}(q, w) \neq \mathcal{A}_{\mathcal{H}}(q', w)$.

To mitigate the state explosion problem inherent in the classical W-method, they extend the approach using random walks as a heuristic. Their approach relies on constructing a test suite using the characterization set $W$, and a state cover set $Pref$; which contains the input prefixes required to reach every state in $\mathcal{A}_{\mathcal{H}}$.

---

[5] We write $row(s)$ to denote the vector of table entries for string $s$ across all columns in $E$, *i.e.*, $row(s) = O_T(s, e)_{e \in E}$

[6] In the context of RNNs, most abstraction refinement methods use a combination of active and passive learning, which we regard as a form of hybrid learning and discuss in the next section.

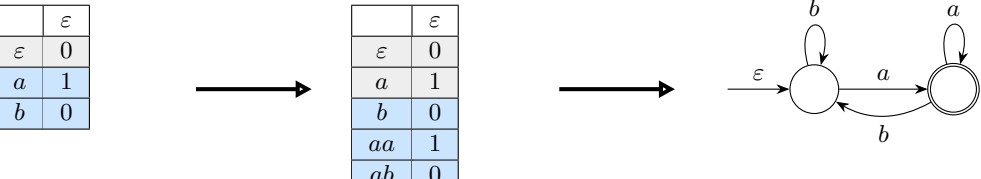

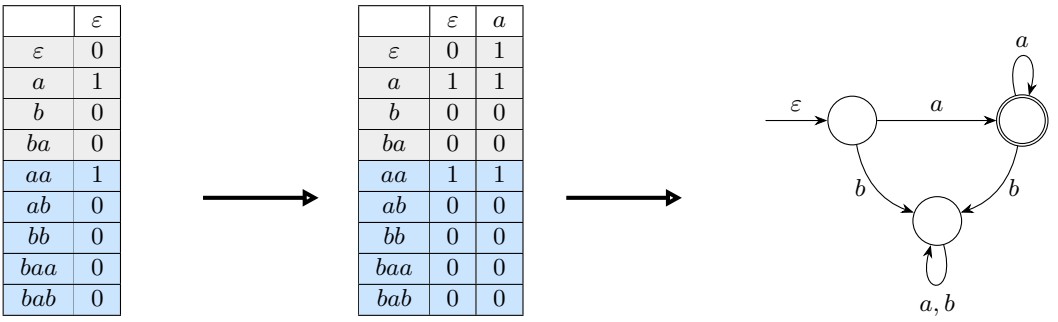

Figure 4: Illustration of the L* algorithm

Their algorithm iterates through the prefixes in the state cover set. At each step, it combines these prefixes with sequences from $W$, while also incorporating random sampling to enhance coverage. Beyond standard benchmarks, the practical efficacy of this approach was substantiated by its performance in the TAYSIR competition (Eyraud et al., 2023), where it secured first place (Muškardin et al., 2023).

### 3.3 Hybrid Learning

The hybrid learning approach, illustrated in Figure 5, integrates active and passive learning through a Counterexample-Guided Abstraction Refinement loop. Initially, the system generates two automata: an actively learned hypothesis $\mathcal{A}_{\mathcal{H}}$ and a passively learned hypothesis $\mathcal{A}'_{\mathcal{H}}$. During the equivalence query phase, the two automata are compared to identify a word $w$ such that $\mathcal{A}_{\mathcal{H}}(w) \neq \mathcal{A}'_{\mathcal{H}}(w)$. If found, this sequence serves as a counterexample and is validated against the RNN to determine the ground truth. If $\mathcal{A}_{\mathcal{H}}$ is inconsistent with the RNN, the CEX is incorporated into the active learner's observation table to reconstruct the hypothesis. Conversely, if $\mathcal{A}'_{\mathcal{H}}$ is incorrect, the passive learner refines its model by splitting the state clusters responsible for the misprediction. This iterative refinement continues until the hypotheses converge and no further counterexamples are identified.

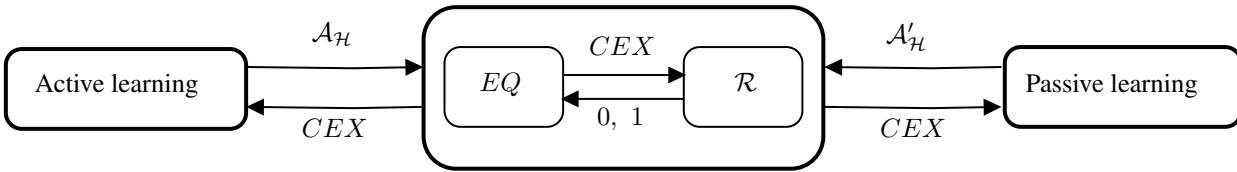

Figure 5: Hybrid learning by counterexample-guided abstraction refinement

Hybrid learning can be interpreted as a mechanism for bringing together underapproximation and overapproximation. While active learning (with an approximation method as the teacher) typically yields minimal automata that underapproximate the target language, passive learning (based on a partition of the states of the RNN, and a support-vector-based refinement of this partition) produces abstracted models that serve as overapproximations. Hybrid learning thus seeks a convergence point between these two.

In practice, passive learning may be based on k-means instead of Support-Vector, as described above, but in that case, $\mathcal{A}'_\mathcal{H}$, does not strictly overapproximate the entire RNN, rather, it overapproximates a specific subset of the RNN 's language, $i.e.,$ $\mathcal{L}(\mathcal{R}) \setminus \mathcal{L}(\mathcal{A}'_\mathcal{H}) \neq \emptyset$. Consequently, achieving convergence between $\mathcal{A}_\mathcal{H}$ and $\mathcal{A}'_\mathcal{H}$ does not provide a formal guarantee of equivalence to the underlying RNN.

This approach was employed by Weiss et al. (2017) for learning FA, followed by their extension (Weiss et al., 2019) for PFA. However, we note that the hybrid learning approach is not employed in any of the tools and applications listed in the following section, since they tend to use either passive or active techniques to learn an automaton from an RNN.

## 4 Applications in Verification, Testing, and Interpretability

Various applications of automata learning from recurrent neural networks have been proposed, most notably in formal verification, model-based testing, and interpretability. In this section, we select and examine papers that have proposed distinct and effective solutions. Our aim is to achieve a balance in this review between breadth and depth, allowing us to capture the full scope of the field and its relevance to bringing trust in neural networks, as well as the essential methodologies employed, their practicality, and limitations.

### 4.1 Formal Verification

The goal of formal verification is to determine if an RNN $\mathcal{R}$ satisfies a property $\mathcal{P}$. One common property of interest is local robustness against adversarial perturbations (Katz et al., 2017; Lemesle et al., 2025), in other words, providing a mechanized mathematical proof that, given an input $x \in D$, adding some small noise $\epsilon \in \mathbb{R}^{d_{in}}$ does not affect the neural network decision, $i.e., \mathcal{R}(x + \epsilon) = \mathcal{R}(x)$.

Previously proposed solutions span from techniques relying on abstractions (Ryou et al., 2020; Tran et al., 2023), to logic frameworks (Jacoby et al., 2020). However, most of these approaches rely on network unrolling, and due to the presence of gating mechanisms and loops in recurrent architectures their verification with such techniques is computationally intensive, limiting the scalability and the precision of the solution. In this section, we discuss not only how automata learning can be used to ease RNN robustness verification, but also how it can be employed to tackle properties beyond adversarial robustness.

Khmelnitsky et al. (2022) propose a Property Directed Verification approach that relies on active learning. We illustrate their approach in Figure 6. Their idea is that, given a specification $\mathcal{S}$ (encoded as a regular language) and an RNN $\mathcal{R}$, the $L^*$ learner constructs a hypothesis automaton $\mathcal{A}_\mathcal{H}$ by asking membership queries to $\mathcal{R}$. Then, instead of answering an equivalence query, they check if the language of the learned hypothesis is a subset of the language of the specification, $i.e., \mathcal{L}(\mathcal{A}_\mathcal{H}) \subseteq \mathcal{L}(\mathcal{S})$. This would lead to proving or disproving the property expressed by $\mathcal{S}$, or finding a counterexample and sending it back to the learner to reconstruct the hypothesis and restart the process.

If satisfied, they proceed to verify the answer by checking if the language of the RNN is a subset of the language of the learned hypothesis, $i.e., \mathcal{L}(\mathcal{R}) \subseteq \mathcal{L}(\mathcal{A}_\mathcal{H})$. To do this, a statistical model checking technique is used to generate test words from a probability distribution and see if they belong to both languages, this step leads either to finding a counterexample which indicates that $\mathcal{L}(\mathcal{R}) \subseteq \mathcal{L}(\mathcal{A}_\mathcal{H})$ is not true, or to validating it under all test words which gives a PAC guarantee of its correctness. If that is the case, the property is satisfied, if not, that means $\exists w \in \mathcal{L}(\mathcal{R}) \setminus \mathcal{L}(\mathcal{A}_\mathcal{H})$, and in this case, $w$ is a counterexample that will be sent back to $L^*$.

If $\mathcal{L}(\mathcal{A}_\mathcal{H}) \subseteq \mathcal{L}(\mathcal{S})$ is not satisfied, it means $\exists w \in \mathcal{L}(\mathcal{A}_\mathcal{H}) \setminus \mathcal{L}(\mathcal{S})$, which is either because the learned hypothesis is incorrect or because the property is violated, and by checking if $\mathcal{R}(w) \neq \mathcal{A}_\mathcal{H}(w)$ and finding it to be the

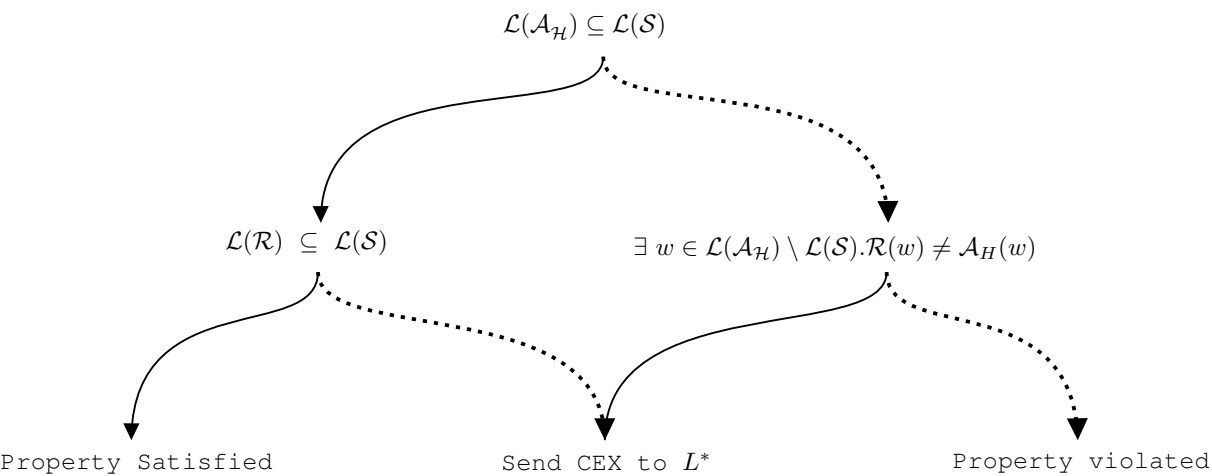

Figure 6: The Property Directed Verification approach, where filled and dotted lines mean the formula evaluates to true and false, respectively.

case, they can confirm that the learned hypothesis is incorrect, then send the counterexample to $L^*$, or if $\mathcal{R}(w) = \mathcal{A}_{\mathcal{H}}(w)$ then the property was violated and $w$ is a counterexample indicating that the specification $\mathcal{S}$ is unsatisfied.

Their experiments suggest that their approach outperforms pure statistical model checking, as well as the model checking of automata overapproximating the RNN semantics (which usually leads to spurious examples since found counterexamples may belong to the language of the RNN abstraction but not to the RNN language itself). The Property Directed Verification approach can prove/disprove a property or find shorter counterexamples, much faster than the other techniques. However, it requires crafting a specification, a non-trivial problem when it comes to neural networks, which makes the solution very limited to a small set of problems that can be expressed in regular languages.

Vengertsev & Sherman (2020) explore the efficiency of *Monte Carlo Model Checking* (MCMC) for the verification of safety properties in RNNs. MCMC methods have demonstrated some effectiveness in the verification of stochastic systems, and given the large space of possible RNN configurations, it can be a viable solution. They use a Labeled Transition System rather than an automaton as the data structure that simulates the evolution of RNN states over time[7].

The LTS states store RNNs hidden states and their associated outputs, and the transitions are labeled by the input vectors. For example, when the RNN reads the following sequence $\mathcal{X} = (x_1, x_2 \ldots x_n)$, the LTS encodes it as follows: $(\mathcal{H}[:0], \mathcal{Y}_0) \xrightarrow{x_1} (\mathcal{H}[:1], \mathcal{Y}_1) \ldots \xrightarrow{x_n} (\mathcal{H}[:n], \mathcal{Y}_n)$ where $\mathcal{H}[:i]$ (respectively $\mathcal{Y}[:i]$) are the hidden states of the RNN (respectively its outputs) after reading the sequence $x_1, x_2 \ldots x_i$, *i.e.*, $\mathcal{H}[:i] = (h_0, h_1 \ldots h_i)$. The LTS states $(\mathcal{H}, \mathcal{Y})$ are then enriched with the following predicates:

◇ **High confidence:** measures how confident each LTS state's decisions are by computing the estimation $0 \leq P(\mathcal{Y}) \leq 1$ ($P$ is given by a softmax, or similar function) of an RNN output sequence $\mathcal{Y}$. The high confidence property $Hi(a)$ holds if $P(\mathcal{Y}) \geq a$, where $0 \leq a \leq 1$ is a hyperparameter.

◇ **Low confidence:** is a similar predicate $Lo(b)$ that applies the same logic by examining if $P(\mathcal{Y}) \leq b$.

◇ **Robustness:** determines if flipping a transition from $q \rightarrow q'$ after reading some input, to $q \rightarrow q''$, requires adding at most a value $r$ to the input.

◇ **Coverage:** is the rate of activated neurons $H$ computed as $\frac{\|H \geq z\|}{dim(H)} \geq c$, given the hyperparameters $c$ and z.

---

[7]The behavior model is thus not learned, and its number of states may be infinite. However, we included this work in our survey since the verification method can be applied to a learned model as well.

Afterwards, Vengertsev & Sherman (2020) define *safety state properties* as Linear Temporal Logic (LTL) formulas built on these predicates: for example, the High Confidence safety state property is defined by the LTL formula $\mathbf{G}Hi(a)$, where $0 \leq a \leq 1$ is a hyperparameter and $\mathbf{G}$ is the temporal operator 'always (**G**lobally)'. They also define two *temporal safety properties*: long-term relationship and memorization. **Long-term relationship** checks if an RNN trained over sequences of length $T$ can also have comparable confidence measures on sequences of length $T + \rho(T)$, where $\rho(T) > 0$ is a linear function. In other words, each sequence of length $T + \rho(T)$ is split in two (at the first $T$ states) and we check that the high confidence property $Hi(a)$ holds at least $u$ times on the first, and property $Hi(d)$ holds $v$ times on the second, where $0 \leq d \leq a \leq 1$, and $u$, $v$ are hyperparameters. Similarly, **Memorization** checks if for each sequence of size $T$, the property $Hi(e)$ where $e = 1 - \epsilon$ is a hyperparameter close to 1 holds at least $m$ times ($0 < m < T$ is also a hyperparameter).

Then, they use Monte-Carlo sampling, with *a posteriori* probabilistic evaluation, to evaluate the probability of each property being true, given a set of randomly generated samples. Their method was evaluated on two toy LSTMs trained for next character prediction tasks. To test their hypothesis that MCMC is effective for RNN verification, they establish the ground truth by an exhaustive search over the LTS to verify each property, then determine the number of Monte-Carlo samples needed to find the correct probabilities and compute the value of the convergence rate. Although the experiments are preliminary, they suggest the relevance of this technique for verification of RNN properties, and they also indicate that temporal safety properties are less efficient to compute as they require significantly more sampling.

Wang et al. (2018) define an edit distance approach to determine the perturbation distance $\gamma$ over the sequences. Their edit distance is an extension of the average edit distance definition, and is computed as follows: $\mathcal{D}(\chi_y, \chi'_y) = \frac{1}{2} \lim_{N \to +\infty} (\frac{\mathcal{D}_y^N}{|\mathcal{X}_y^N|} + \frac{\mathcal{D}_{y'}^N}{|\mathcal{X}_{y'}^N|})$. Given an FA $\mathcal{A}$ we have $\chi_y = \{x \mid \mathcal{A}(x) = y\}$, $\mathcal{D}_y^N$ sums up the minimum edit distance added to a string $x \in \mathcal{X}_y^N$ to flip its label from $y$ to $y'$ (likewise $\mathcal{D}_{y'}^N$). The FA is inferred using passive learning from a trained RNN, and the learning step relies on k-means clustering. The FA is treated as an oracle that is capable of recognizing, with a high fidelity, the accepted sequence by the RNN. Their proposed algorithm initializes a randomly sampled dataset of $N$ strings $\mathcal{X}$ accepted by the RNN and its learned FA, adds perturbations to the strings of each sequence to push it to the decision boundary (by adding the maximally allowed perturbation distance $d$), then loops over all sequences and calculates the total number, *count*, of sequences misclassified by the RNN after the added perturbations at some string $x_i \in \mathcal{X}$. The adversarial robustness is then measured by taking $\gamma = 1 - \frac{count}{N}$.

Mayr et al. (2021) combine $L^*$ with PAC learning (Section 3.2) to sample a dataset which guarantees that the learned hypothesis is $1 - \epsilon$ correct with a confidence $1 - \delta$. Then, applying language inclusion techniques and model checking, the automata can be used to verify RNNs against predefined properties (mostly restricted to regular languages) while providing theoretical, probabilistic guarantees of correctness.

The papers discussed in this section are summarized in Table 1, which presents information regarding:

**Method:** the learning style and the learned model, the kind of RNN this method is applied to, and the kind of properties that can be verified.

**Applications and their scale:** we consider three broad categories of scales. Algorithms extracting automata from toy models trained on formal languages. Algorithms extracting automata from models trained on real-world data (such as MNIST and IMDB). Algorithms extracting automata from large models whose parameter count is more than $10^6$.

**Code availability:** when the code is publicly available, we provide a link to its repository.

Table 1: Summary of RNN verification approaches

| Reference | Learning Style | Learned Model | RNN Type | Applications | Scale | Verification Method | Property Verified |
|---|---|---|---|---|---|---|---|
| (Khmelnitsky et al., 2022)[1] | Active | FA | L | Regular lang. | 1 | Statistical MC | Specification FA |
| (Wang et al., 2018) | Passive | FA | E, L, G | Regular lang. | 1 | Edit distance | Robustness |
| (Mayr et al., 2021) | Active | FA | L | Regular lang. | 1 | Model checking | Robustness |
| (Vengertsev & Sherman, 2020) | — | LTS | L | Natural lang. | 2 | Monte Carlo MC | LTL properties |

**RNN type**: E: ERN, L: LSTM, G: GRU.
**Scale:** 1: synthetic/formal languages; 2: real-world data; 3: larger models (more than $10^6$ parameters) and/or complex datasets.
[1] `https://github.com/LeaRNNify/Property-directed-verification`

### 4.2 Model-Based Testing

Recurrent neural networks are often trained to solve tasks that require the ability to express human-like cognitive capacities, such as speech synthesis, language understanding, etc. Hence, crafting specifications to formally verify these networks against desired properties is a very challenging task. Model-based testing offers an alternative solution that systematically searches for cases in which the property under test fails, which is practically relevant for characterizing neural network possible failures. However, while relying solely on testing methods, we cannot ensure that the neural network is certified, simply because testing demonstrates the presence of failures, not their absence. Nonetheless, this technique remains valuable to quantitatively analyze neural network behaviors and guide their improvements, as we will show in this section.

DeepStellar (Du et al., 2019) relies on model-based testing for robustness analysis. They use passive learning to extract a DTMC $\mathcal{A}$. To enable systematic testing, *i.e.,* test set sufficiency and input redundancy, the authors define trace similarity and coverage criteria metrics based on the extracted automaton $\mathcal{A}$. Trace similarity metrics are designed to measure, given two input sequences $\mathcal{X}$ and $\mathcal{X}'$, to what extent they lead to the same RNN behavior, *i.e.,* the states (or transitions) triggered while propagating $\mathcal{X}$ through $\mathcal{A}$ are also triggered during propagation of $\mathcal{X}'$. This is defined by calculating the total number of abstract states (respectively, abstract transitions) triggered by $\mathcal{X}$ and $\mathcal{X}'$ divided by the total number of states triggered by $\mathcal{X}$ or $\mathcal{X}'$. Coverage criteria are metrics to measure data sufficiency in assessing RNN global behavior. Six metrics were proposed to measure state and transition coverage, namely: *Basic*, *n-Step Boundary*, and *Weighted* state coverage, and transition coverage metrics defined in a similar way.

Given a test dataset $D_T$, *basic state coverage* is defined by calculating the number of states in $\mathcal{A}$ that are visited while propagating the sequences from $D_T$ divided by the total number of states in $\mathcal{A}$. This can be problematic in the case where $D_T$ triggers states that are not defined in $\mathcal{A}$, since the DTMC is constructed from the training data and their passive learning algorithm does not guarantee equivalence or overapproximation of the RNN behavior. To address this, the *n-Step Boundary coverage* extends the corners of the partitioned abstract state space by creating new regions that have at most a distance n between them and their closest abstract states in $\mathcal{A}$. See example in Figure 7.

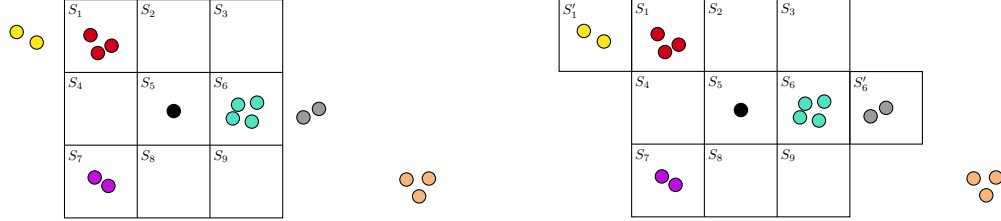

Figure 7: n-Step State Boundary Coverage (n = 1)

Then the n-Step boundary coverage is calculated by taking the ratio of the total number of states in the boundary region, that were visited during the execution of the test set by the total number of states in the boundary region. *Weighted state coverage* is another metric that extends the Basic state coverage by attributing weights to the states to quantify their importance. In other words, an abstract state in $\mathcal{A}$ that contains $p$ hidden states has an $m$-value, while a state that contains $p'$ hidden states such that $p' << p$ is of less importance. As shown in their experiments, the weighted coverage criteria perform better in capturing erroneous RNN behaviors. Basic and Weighted coverage criteria are defined on the transition level in a similar fashion.

The authors apply these metrics to two tasks: adversarial sample detection and robustness testing. For adversarial detection, the authors propose to initialize 3 datasets: a reference dataset $D_R$ that contains arbitrary examples, $D_B$ for benign examples, and $D_A$ for adversarial examples. They apply their trace similarity metrics to compare the threshold distances between benign and adversarial inputs, and then use the distances to train a linear regression classifier for adversarial sample detection. For the experiments, LSTMs and GRUs were trained for MNIST digits classification and speech recognition. The RNNs are not toy examples; some of them have more than $1.2 \times 10^8$ parameters, which is adequate to assess the practicality of the technique. Results indicate a weak to moderate correlation between trace similarities and prediction differences.

For robustness testing, DeepStellar uses coverage criteria to guide the generation of a test suite. Their algorithm selects inputs from an initial test queue, then randomly perturbs them, which creates from that initial input a set of new samples. An input from the new samples is then picked and passed through the RNN to collect its corresponding hidden states. If the predictions fail, then the input is added to the failed test set. Otherwise, if it leads to better coverage scores, they add it to the initial test suite to be perturbed next. The algorithm is tested on MNIST classification using LSTMs and GRUs, and it demonstrates the potential of the coverage criteria in guiding the generation of test suites, as it outperforms random sampling.

DeepCover (Golshanrad & Faghih, 2024) uses k-means clustering to learn a finite state transducer[8], rather than the grid-based partitioning used in DeepStellar (Du et al., 2019). Since we need to set a priori the number of centroids for k-means, the learning algorithm chooses $k$ after evaluating the quality of the learned automata. They define metrics to assess the quality of the learned FA, mainly *Goodness* and *Scale*, which they use later on during the passive learning stage by defining thresholds on these scores and seeking the smallest $k$ satisfying such a threshold.

Goodness is measured by combining two scores, Purity and Richness, in a linear function. Purity is the ratio of the maximum number of concrete states that have the same label in a given abstract state, to the total number of final concrete states in all abstract states (final states are states whose labels are non-Null). Higher Purity scores reflect higher fidelity to the original RNN, since, if the final states in the automaton are correctly labeled, Purity is high, and if there is a confusion or a mismatch in labels, Purity is low. Richness is the ratio of the number of concrete states with non-Null labels to the total number of abstract states that contain non-Null concrete states. In other words, it tells us how many of the concrete states with the same label are grouped under the same abstract state. The higher the richness is, the more effective the learned automaton is in capturing the RNN behavior. Scale is the ratio of the total number of abstract states containing final concrete states (that is, those with labels) to the total number of labels. Higher scale scores indicate higher sparsity in the cluster space.

Their results demonstrate that the learned automata have slightly larger sizes compared to (Du et al., 2019), but exhibit better fidelity to the original models, as demonstrated by the Goodness scores. This suggests that the clustering approach plays a significant role in the quality of the learned model. Similarly, they define a set of coverage metrics in the same fashion as in (Du et al., 2019), however, they focus more on the final concrete states rather than the whole state space. This label awareness focus, as demonstrated in the experiments, provides a better characterization of underexplored states targeting RNN robustness.

---

[8]The motivation behind using a transducer rather than a semiautomaton lies in the need to reproduce the RNN input-output behavior.

On the other hand, Marble (Du et al., 2020) follows a refinement approach that continuously refines the learned automaton for higher fidelity. Their approach is founded on a pointwise robustness estimation technique, which is based on approximating the likelihood that flipping some symbol in an input word doesn't change its path in the automaton. Unlike the previous technique, Marble builds an MDP by partitioning the states that have similar robustness scores. This would eventually lead to obtaining abstract states where transitions with the same input may lead to different successor states. After creating the MDP to estimate the probabilistic input-output behavior of the RNN, they iteratively refine its state abstraction by splitting the partitions that do not have similar robustness measures. The latter step is theoretically guaranteed to converge since the space of concrete states is finite.

While these approaches focused on the robustness of RNNs, DeepMemory (Zhu et al., 2021) proposes a method to analyze memorization. Given a trained neural language model and its training data, they create a first-order Markov model to quantify the memorization probability rates of input sequences following 3 steps. They start by constructing a *memorization-analysis oriented model* by creating a transition system from the concrete states and their transitions, then to abstract the states, they use PCA dimensionality reduction followed by DBSCAN (Ester et al., 1996) clustering. The PCA step is combined with Relative Information Loss (Geiger & Kubin, 2012) to make sure the inputs are reduced to the lowest dimension $k$ without severe loss of information. Next, they define a metric that computes how likely a sequence is to be memorized, given an input sequence $\mathcal{X}$, its prefix, and suffix from the training data. More intuitively, if the RNN $\mathcal{R}$ produces the same output for the suffix after processing a prefix sequence, then the prefix is likely memorized. This metric is then applied to classify the concrete states (respectively transitions) as memorization and non-memorization states (respectively transitions). Finally, a first-order Markov model is constructed by computing the memorization probabilities of abstract states (respectively transitions). The final model can then be applied to data leakage risk assessment by checking if sequences from the test data have high memorization probabilities (which may indicate that these sequences exist in the training data and potentially lead to data leakage issues). They also suggest that their method can be useful in assisting in dememorization, by mutating the detected potentially memorized sequences. The experiments confirm the utility of their approach in analyzing LSTM-based neural language models trained on relatively large datasets (1M to 4M training sentences). However, their automata construction suffers from severe scalability issues as it requires up to 22 hours to create the model.

We summarize the discussed works in this section in Table 2. We also include a scale assessment and links to available code.

Table 2: Summary of RNN testing approaches

| Reference | Learning Style | Learned Model | RNN Type | Applications | Scale | Property Under Test |
|---|---|---|---|---|---|---|
| (Du et al., 2019)[1] | Passive | DTMC | L, G | Speech, MNIST | 3 | Adversarial resilience |
| (Golshanrad & Faghih, 2024)[2] | Passive | FA | E, L, G | Speech, MNIST | 2 | Erroneous behavior |
| (Du et al., 2020) | Passive | MDP | L, G | Natural lang., speech | 2 | Adversarial resilience |
| (Zhu et al., 2021) | Passive | Markov model | L | Natural lang. | 2 | Memorization |

**RNN type**: E: ERN, L: LSTM, G: GRU.
**Scale:** 1: synthetic/formal languages; 2: real-world data; 3: larger models (more than $10^6$ parameters) and/or complex datasets.
[1] https://github.com/xiaoningdu/deepstellar
[2] https://github.com/pouriagr/deep-cover

## 4.3 Interpretability

Recurrent neural network interpretability is the ability to provide explanations that can be understood by humans on how these models operate internally. A dominant paradigm for achieving this is the learning of an underlying automaton, as an interpretable surrogate. While the literature is rich with learning methodologies of quantitative (Dong et al., 2020; Zhang et al., 2021; Ayache et al., 2019; Eyraud & Ayache, 2024) and

abstract (Xu et al., 2021; Wang et al., 2023) automata for RNN explainability, we focus here on recent advancements that utilize these automata to derive more profound insights.

Kaadoud et al. (2022) uses automata learning as a global explanation method to interpret a reinforcement learning agent's strategy evolution during training. They use passive learning to model the behavior of the agent trained to solve a reasoning problem: the Tower of Hanoi. After generating a dataset describing its input-output behavior, they use the data to train an LSTM model. Passive learning based on clustering is applied to learn automata representations from the LSTM describing the agent's strategy evolution during training. A key contribution of their method is its ability to explain an agent's *Aha moment*, *i.e.,* the point at which the knowledge representation stabilizes and the agent stops learning. As the agent training is performed over $n$ episodes, automata can be learned after each episode and compared to see the moment at which the automaton does not change between episodes $i$ and $i+1$. At that stage, we know that the agent strategy is represented by that automaton.

While Kaadoud et al. (2022) applied automata learning as merely a visualization method, other works went beyond that. RNNRepair (Xie et al., 2021) presents a method to understand and correct RNN misbehavior by *fault localization* and *remediation*. Their method is based on passive learning via clustering to learn an automaton from a trained RNN (using the training dataset). The automaton is then used to perform *influence analysis* and identify the most influential training samples of a given input sequence (or its segment) from test samples. Based on the influence analysis, they developed a debugging mechanism to identify and offset test errors. For the clustering step, they apply Gaussian Mixture Models, however, since it requires a predefined number of clusters $k$, they propose a technique to find the best value for $k$, called a *state stability* metric. The state stability metric is a way to compute the dominant number $\zeta$ of concrete states having the same label in a given abstract state. Then $k$ is minimized such that $\zeta$ is greater than a hyperparameter $\theta$. Higher $\theta$ values lead eventually to a more concentrated behavior in the states, or what they refer to as $\zeta$-stability. RNNRepair detects and debugs states that led to a misclassification by first localizing the fault using influence analysis, then remediating it using influence scores.

Influence analysis is achieved through an *influence function* $\mathcal{I}(q_{i-1}, x_i)$ that groups all the inputs that influenced[9] the decision taken while transitioning from state $q_{i-1}$ after reading $x_i$.

To define the influence score, we first need to introduce *state confidence* $\mathcal{C}$, which measures the ratio of the maximum number of concrete states with the same label to the total number of concrete states in a given abstract state. Given a sequence $\mathcal{X}$ and its path $\pi_\mathcal{X} = q_0 \xrightarrow{x_1...x_n} q_n$, the temporal feature of the sequence is $\mathcal{F}_\mathcal{X} = (f_0 \dots f_n)$ where $f_i = (i, \mathcal{C}_{q_i}, \mathcal{Y}_{\hat{h}_i})$ is a tuple containing the identifier of the state, its confidence score, and the RNN prediction of its abstract state $\hat{h}_i$. Finally, the influence score is defined as $\text{InfluenceScore}(\mathcal{X}_{train}, \mathcal{X}_{test}) = \text{Similarity}(\mathcal{F}_{\mathcal{X}_{train}}, \mathcal{F}_{\mathcal{X}_{test}})$, where Similarity is a distance measure, such as the $l_p$ norm.

When the misclassification is caused by 1 or multiple segments in the input, they determine the input's set of segments that led to the error $\{x_i | 1 \le i \le n \land \mathcal{I}(q_{i-1}, x_i) < \gamma\}$ for a given hyperparameter $\gamma$. Now, for remediation, they randomly sample a set of training sequences that contain the misclassified segments while ensuring they lead to the correct prediction by computing their influence score. The new samples are then added to the training set to restart the RNN training.

The same principle is followed to localize and remediate misclassification caused by the input sequence by first finding the set of samples in the training data that are misclassified by the model and using data augmentation to enrich the training set, hence guiding the model towards the correct prediction associated with these sequences. As demonstrated by their benchmarks, the fault localization and remediation method is about one order of magnitude more efficient than random data augmentation.

Ishimoto et al. (2023) uses a k-means clustering technique from Dong et al. (2020) to learn a PFA from an RNN, and then applies a fault-localization technique to the earlier. This technique consists of computing

---

[9]The most representative training samples that led to that prediction.

a *suspiciousness score* for each $n$-gram (a sub-sequence of length $n$) of an input sequence $x$. That score indicates the extent to which the $n$-gram is likely responsible for the misbehavior of the RNN, while its average can be used to select the data samples that are likely to fail to be predicted. The authors then experiment on several datasets and compare their results to SoTA tools. The experiments show that their method statistically outperforms SoTA Fault Localization techniques, but takes a longer execution time. The experiments also show that the two parameters of their method, $k$ and $n$, can be optimized for each dataset.

Wei et al. (2024) tackle the problem of explaining RNNs trained for Natural Language Processing (NLP) tasks. They rely on passive learning via clustering to learn a WFA. During the learning process, they identify a few issues, such as data sparsity and context loss.

For the former, the authors use Zipf's law[10] (Powers, 1998) to mathematically demonstrate the severity of the transition sparsity problem, they estimate that the median word frequency in natural language datasets is extremely low, often $\approx 2$ transitions per state. To address this issue, they propose data augmentation techniques alongside the *missing row complement* method that leverages semantic distances between abstract states. Specifically, for sparse abstract transitions, they simulate the transition behaviors from similar abstract states, weighted by their semantic distances to ensure that states with similar behavior contribute to filling in the transition gaps.

RNNs, particularly LSTM architectures, are widely utilized in NLP tasks for their capacity to maintain temporal dependencies across extended sequences. To faithfully represent this behavior, a learned automaton must account for such long-range context. To this end, the authors propose a *context-awareness enhancement* mechanism designed to augment the WFA's ability to retain information from preceding states during input processing. This method introduces a *retention rate* hyperparameter $\alpha$, which incorporates self-loops into the WFA state transitions. By incrementing the diagonal elements of the transition matrices by $\alpha$, the automaton can preserve a portion of the current state information during transitions, allowing the WFA to better capture and preserve information flow.

Table 3 presents a brief summary of the papers presented; this table also includes links to code repositories of the works discussed herein.

Table 3: Summary of RNN interpretability approaches

| Reference | Learning Style | Learned Model | RNN Type | Applications | Scale | Explanation |
|-----------|----------------|---------------|----------|--------------|-------|-------------|
| (Kaadoud et al., 2022)[1] | Passive | FA | L | RL agent behavior | 2 | Learning stability |
| (Xie et al., 2021)[2] | Passive | FA | L, G | Natural lang., MNIST | 2 | Fault localization & remediation |
| (Ishimoto et al., 2023)[3] | Passive | PFA | E, L, G | Natural & formal lang., MNIST | 2 | Fault localization |
| (Wei et al., 2024)[4] | Passive | WFA | L | Natural lang. | 2 | Decision making |

**RNN type**: E: ERN, L: LSTM, G: GRU.
**Scale:** 1: synthetic/formal languages; 2: real-world data; 3: larger models (more than $10^6$ parameters) and/or complex datasets.
[1] https://github.com/ichraibi/Extracting_Aha_moment_from_Qlearning_agent_through_IKE-XAI_method
[2] https://bitbucket.org/xiaofeixie/rnnrepair
[3] https://github.com/posl/PAFL-replication
[4] https://github.com/weizeming/Extract_WFA_from_RNN_for_NL

---

[10]Zipf's law is an empirical observation in linguistics stating that the frequency of a word in a text corpus is inversely proportional to its rank order. In other words, the most frequent word will occur approximately twice as often as the second most frequent word, and three times as often as the third. In NLP, this heavy-tailed distribution implies that a small fraction of words from the selected corpus accounts for the vast majority of occurrences, while a long tail of rare words appears very infrequently.

## 5   Discussion

In this survey, we have provided a synthesis of the state-of-the-art in automata learning from recurrent neural networks and its applications. The many learning styles (passive, active, or hybrid) are adaptable to white box models, as well as fully black box ones, and result in the construction of surrogate models that can help answer the following questions about RNNs: How to formally prove their robustness, how to test them, and how to understand their behavior. However, we also identified a few limitations that need to be addressed to make this method practically relevant for modern AI safety challenges.

**Automata learning algorithms**   The algorithmic landscape of automata learning, particularly in the context of RNNs faces several challenges. First, passive learning relies on state-merging algorithms that are often not accurate enough; this issue was studied with various merging algorithms (Soubki & Heinz, 2023). Second, active learning, while promising for decoding black box models, remains underexplored and inherently slow when learning languages with large alphabets. While many active learning approaches were based on the $L^*$ algorithm, other algorithms exist and show some potential (Vaandrager et al., 2021; Isberner et al., 2014). Furthermore, answering equivalence queries with formal mathematical guarantees poses a substantial theoretical and practical obstacle. Third, there is a lack of standardized, appropriate benchmarks for neural networks. While most benchmarks in automata learning were conceived to deal with small, finite sets of alphabets, RNN input alphabets can be quite large, and there is no standard benchmark specific to RNNs. As the goal is to evaluate not only the scalability of these algorithms, but their correctness as well, benchmarks for formal grammars over infinite alphabets should be extended and experimented with as in Slimi et al. (2025) or Van Der Poel et al. (2024), so that we can get closer to how these models operate, *i.e.,* on possibly infinite alphabets, then apply them to RNNs trained on complex tasks such as language modeling or time series forecasting.

In addition, alternative approaches for learning automata exist, such as methods based on SAT solvers (Heule & Verwer, 2010; Ulyantsev et al., 2016), and more recently (Dell'Erba et al., 2024; Meng et al., 2025). Techniques based on machine learning are also emerging, for instance Chen et al. (2026) and Vazquez-Chanlatte et al. (2025) rely on Large Language Models in an active learning framework. Hosseinkhani & Leucker (2025) leverage reinforcement learning (Q-learning) for passive learning. Such alternative solutions can offer a fresh perspective on the field and may serve in one way or another in advancing it further, whether by providing theoretical guarantees of learning or heuristics that speed up computation and help scale to larger models.

**Formal verification**   The mathematical rigor behind formal verification requires soundness of the analysis: if we want to prove a property holds for an RNN $\mathcal{R}$ by using verification techniques, *e.g.,* model checking, on the surrogate model $\mathcal{A}$, we must have the equivalence of languages $\mathcal{L}(\mathcal{R}) \equiv \mathcal{L}(\mathcal{A})$. Neither passive learning nor active learning provides such a formal guarantee in general. While RNNs can be more expressive than finite automata, it is possible to bound the length of the processed sequences and approximate it with an FA, however, this would limit the verification scope as it will only be capable of proving properties over regular languages. Proving properties beyond regular languages remains an open problem.

More recent works on RNN verification (Lin et al., 2026; Choi et al., 2025) build heavily on abstraction techniques. Relying solely on abstraction is not efficient in the case of RNNs due to their complexity, *i.e.,* the gating mechanism and the recurrence, which lead to growing abstractions after each iteration, resulting in very large (hard to compute) and imprecise solutions. This is evident in the experimentation sections, which illustrate a challenge to go beyond small network architectures. While automata learning does not provide soundness guarantees, it can be applied as a heuristic to help the verifier find tighter abstractions. As in Glunt et al. (2025), some computation can be simplified by functional decomposition tricks. We believe that automata learning can be an efficient way to automate such decomposition and eventually assist the verifier during the verification process.

**Model-based testing**   The model-based testing approach has been more successful with RNNs than formal verification. Yet, the success is relative to robustness analysis, specifically the evaluation of model resilience against adversarial perturbations, and other properties remain underexplored. Memorization is of particular

interest when it comes to recurrent networks. Given that RNNs are architecturally designed to simulate memory through hidden states, they are susceptible to merely memorizing specific training sequences instead of learning generalizable patterns. Thus, we need to ensure they do not pose privacy and data leakage issues by certifying that they do not memorize the training/user data. Model-based testing of RNNs through automata learning could also benefit from more scalable and precise learning algorithms capable of modeling the RNN capacity to process sequences. For instance, most papers we reviewed build their algorithms on top of passive learning, such approaches may benefit from the hybrid learning style for more precision, *i.e.,* better fidelity to the original RNN, and it may be better suited for coverage measure since it combines the two learning styles in a more efficient counterexample generation strategy.

**Interpretability**   Numerous works aimed at enhancing RNN interpretability conclude their analyses upon learning the automaton. The claim is that the automaton itself serves as a sufficient explanation for the model's behavior, which may be a valid perspective given that automata are inherently more interpretable structures. However, this perspective limits the possibilities for better solutions, in two ways. First, for the learned automaton to serve as an effective interpretability tool, it must remain compact. The comprehensibility of an automaton is severely impacted by its structural size. This is determined not only by the number of states, but also by the density of the connections between them. A dense graph with few states can be just as difficult for a human to interpret as a sparse graph with many states. Therefore, evaluating the quality of the explanation needs to take into account the number of states as well as the transitions. Second, it often overlooks the potential for deeper insights and a more comprehensive understanding of the underlying model dynamics, *i.e.,* while automata learning offers a way to visualize models internal behavior, it does not efficiently *explain* why, or why not, a decision is taken instead of another.

To better situate automata-based interpretability, it is helpful to compare it with two prominent alternative paradigms for explaining sequential models: *attribution-based* and *probing-based* methods. Attribution-based methods assign importance scores to individual input features or time steps for a given prediction. They operate at the instance level, producing local explanations that highlight *which* parts of an input sequence most influenced a particular output. While these methods scale well to large models and are useful for debugging specific predictions, they do not reveal the model's internal state dynamics or its global behavior. Moreover, for sequential models, attribution methods often struggle to capture the temporal dependencies and stateful interactions that fundamentally characterize RNNs. Probing-based methods take a different approach by training auxiliary classifiers on the internal representations of a neural network to determine what information (*e.g.,* syntactic structure, grammatical properties) is encoded at various layers or time steps. While probes can reveal *what* a model knows, they provide correlational rather than causal evidence; the presence of information in a representation does not guarantee that the model actively uses it for its predictions (Hewitt & Liang, 2019). Furthermore, probes offer no executable model of behavior and cannot be directly leveraged for downstream tasks such as testing.

In contrast, automata-based methods produce a global, structural, and inherently interpretable model that captures the stateful dynamics of the RNN across its entire input space rather than for individual instances. Additionally, automata as explanations are easier to compute compared to standard explanation methods, as discussed by Lin et al. (2024) for the generation of counterfactual explanations for sequential models (such as RNNs and Transformers), and Marzouk et al. (2025) for computing Shapley additive explanations. However, automata-based approaches face their own challenges, as discussed above. These paradigms are therefore best viewed as complementary: attribution and probing methods provide fine-grained, local or representational insights, while automata learning provides a global, mechanistic, and formally grounded account of model behavior.

**Beyond standard recurrent models**   We identify here two main limitations: keeping up with modern recurrent models, and the ability to handle hybrid architectures—combining recurrence with other mechanisms, such as a convolution. The vast majority of the methods mentioned in this review focused on standard recurrent models, *i.e.,* simple and gated (LSTM, GRU) recurrent networks. Nonetheless, modern recurrent variants are being actively developed and shown to perform orders of magnitude better than standard ones. Such architectures include selective state space models such as Mamba (Gu & Dao, 2024), offering fast RNN-like inference combined with Transformer-like efficient training. Another architecture, which can be

seen as a hybrid between Transformers and recurrence, was proposed by Peng et al. (2023). Many proposed works perform automata extraction from Transformers (Song et al., 2024; Adriaensen & Maene, 2025), and given that modern RNNs are close to transformers characteristics, these results suggest a promising avenue for automata extraction from those models as well. Also, RNNs are sometimes combined with convolutions (Bradbury et al., 2017) to enhance their performance in some task-oriented applications. For modern RNN architectures, such as state space models, the applications of automata learning may be straightforward since they do not differ from standard RNNs in terms of operations and structure, *i.e.,* compositions of state and transition functions that play a central role in the algorithmic landscape of automata learning from RNNs. However, scalability to these models is the predominant challenge, given their large size. On the other hand, hybrid models may require novel algorithms to efficiently manage the combination of operations.

## 6 Conclusion

The goals of this review paper were threefold. First, to introduce the problem of automata learning from recurrent networks and its state-of-the-art. Second, to investigate the applications of this method to establish trust in RNNs. Third, to identify the critical limitations from an application standpoint that need to be addressed. We presented different learning styles, which can differ based on the model and data availability. The learned automaton can then be applied as a surrogate model to verify properties such as robustness, test the model against adversarial examples and memorization scenarios, and explain the internal behavior of the RNN in a human-interpretable manner. However, we also identified a few limitations, which we believe are critical to the further advancement of this field. We aimed for a self-contained paper that systematically reviews the latest trends of this method, balancing depth and breadth. We hope our work serves beginners, as well as seasoned researchers who work in this field. With scalable solutions and stronger guarantees, automata-based surrogates can become a unifying basis for certification, testing, and interpretation of recurrent networks.

**Acknowledgments**  We thank the Action Editor and the anonymous TMLR reviewers for the time they dedicated to our work and for their constructive feedback, which helped improve the quality of this paper. This work was supported by the Safe AI through Formal Methods (SAIF) project, funded by the France 2030 government investment plan, managed by the French National Research Agency under the reference ANR-23-PEIA-0006.

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

# A    Notation tables

Table 4: Automata and formal language theory notation.

| Symbol | Meaning |
|---|---|
| $\Sigma$ | Alphabet (finite, non-empty set of symbols) |
| $\sigma, \sigma_i$ | Symbol (letter) in $\Sigma$ |
| $w$ | Word (finite sequence of symbols $\sigma_1 \sigma_2 \ldots \sigma_n$) |
| $|w|$ | Length of word $w$ |
| $\Sigma^*$ | Set of all finite words over $\Sigma$ (Kleene closure) |
| $\mathcal{L}$ | Language (a subset of $\Sigma^*$) |
| $\mathcal{A}$ | Finite automaton |
| $Q$ | Finite set of states |
| $Q_I$ | Set of initial states |
| $Q_F$ | Set of final (accepting) states |
| $q, q_i$ | Individual state in $Q$ |
| $\delta$ | State transition function ($\delta : Q \times \Sigma \to Q$) |
| $\pi_w$ | Path of word $w$ in automaton $\mathcal{A}$ |
| $\mathcal{L}(\mathcal{A})$ | Language recognized by automaton $\mathcal{A}$ |
| $\mathcal{L}(\mathcal{R})$ | Language recognized by RNN $\mathcal{R}$ |

Table 5: Recurrent Neural Network notation.

| Symbol | Meaning |
|---|---|
| $\mathcal{R}$ | Recurrent Neural Network (as a function $\mathbb{R}^{d_{in}} \to \mathbb{R}^{d_{out}}$) |
| $F_\theta$ | State transition function of the RNN, parameterized by $\theta$ |
| $G_\phi$ | Readout (output) function of the RNN, parameterized by $\phi$ |
| $\theta$ | Trainable weight matrix of the state transition function |
| $\phi$ | Trainable weight matrix of the readout function |
| $d_{in}$ | Dimensionality of the input space |
| $d_{out}$ | Dimensionality of the output space |
| $d_h$ | Dimensionality of the hidden state space |
| $x_t$ | Input vector at time step $t$ ($x_t \in \mathbb{R}^{d_{in}}$) |
| $\mathcal{X}$ | Input sequence $(x_1, x_2, \ldots, x_T)$ |
| $h_t$ | Hidden state vector at time step $t$ ($h_t \in \mathbb{R}^{d_h}$) |
| $h_0$ | Initial hidden state |
| $\mathcal{H}$ | Sequence of hidden states $(h_1, h_2 \ldots h_T)$ |
| $\mathcal{H}[: i]$ | Subsequence of hidden states up to time $i$: $(h_0, h_1 \ldots h_i)$ |
| $y_t$ | Output (prediction) at time step $t$ ($y_t \in \mathbb{R}^{d_{out}}$) |
| $\mathcal{Y}$ | Sequence of outputs $(y_1, y_2 \ldots y_T)$ |
| $D$ | Dataset of input sequences $\mathcal{X}$ |

