# OpenReview forum: "Automata Learning from Recurrent Networks: A Critical Synthesis for Verification, Testing, and Interpretability"
_TMLR — Decision pending for TMLR_

### Review · Reviewer_hsbm · 2026-03-12

**Summary Of Contributions:**

The paper presents a survey on automata learning from recurrent neural networks (RNNs) and reviews how automata-based surrogate models can be used to analyze and understand neural network behavior. The authors summarize existing approaches for extracting automata from RNNs, categorizing them into passive, active, and hybrid learning frameworks, and discuss how the learned automata can support tasks such as formal verification, model-based testing, and interpretability. The paper also reviews representative tools and methods in these areas and highlights current limitations and open challenges, including scalability issues, lack of standardized benchmarks, and the difficulty of verifying properties beyond regular languages.


A key strength of the paper is that it provides a structured overview of a relatively specialized research direction that connects automata theory and neural network analysis. The taxonomy of learning methods and the discussion of applications in verification and testing help organize the literature and may be useful for newcomers to the field. However, the paper primarily synthesizes existing work and does not introduce new methods or empirical studies, which somewhat limits its novelty. In addition, the discussion of modern architectures and large-scale models remains relatively brief.

**Additional Comments:**

See requested changes.

**Audience:**

Yes

**Audience Explanation:**

Researchers working on machine learning interpretability, formal verification, neural network testing, and theoretical aspects of deep learning may find this survey useful. The paper connects ideas from automata theory and neural networks, providing a consolidated view of methods that aim to extract interpretable symbolic models from RNNs. While the topic is somewhat specialized, it aligns with ongoing research interests in trustworthy and interpretable machine learning, which are relevant themes for the TMLR audience.

**Broader Impact Concerns:**

I do not see major ethical concerns associated with this work. The paper focuses on improving the interpretability, verification, and reliability of neural networks, which generally contributes positively to the development of safer and more trustworthy AI systems.

**Claims And Evidence:**

Yes

**Claims Explanation:**

The claims made in the paper are generally supported by references to existing literature and a structured review of previously proposed methods. Since the work is primarily a survey rather than a novel technical contribution, the evidence takes the form of summaries of prior studies and reported experimental findings rather than new empirical evaluations. The cited works provide sufficient context for the claims about the applicability of automata learning to verification, testing, and interpretability of RNNs. However, the paper would benefit from more systematic comparisons or synthesis (e.g., clearer quantitative summaries or unified evaluation criteria) to further strengthen the evidential support of its discussion.

**Requested Changes:**

Weaknesses:

1. Limited novelty. The paper is primarily a survey that synthesizes existing literature on automata learning from recurrent neural networks. While the overview is informative, the work does not introduce new algorithms, theoretical analysis, or empirical studies, which somewhat limits the novelty of the contribution.

2. Lack of systematic comparison across methods. Although many approaches are reviewed, the paper does not provide a unified comparison of their characteristics, such as scalability, computational complexity, or fidelity to the original RNN behavior.

3. Limited coverage of modern architectures. The survey mainly focuses on classical RNN architectures (e.g., Elman networks, LSTMs, and GRUs), while more recent sequence modeling architectures are only briefly discussed.

4. Limited empirical illustration. The paper summarizes prior work but does not include illustrative examples or case studies that demonstrate how automata learning can be practically applied to analyze neural network behavior.

5. Scalability challenges are not deeply analyzed. While the discussion section mentions scalability issues, the analysis remains relatively high-level and does not provide concrete technical insights into possible solutions.

Questions:
1. Many of the reviewed approaches are evaluated on relatively small RNN architectures or synthetic tasks. How well do these methods scale to modern large-scale models or real-world sequence modeling tasks?

2. Since automata learning produces surrogate models, how accurately do the extracted automata represent the behavior of the original neural networks in practice?

3. The paper highlights the lack of standardized benchmarks for this research area. What characteristics should such benchmarks include to enable fair comparison of different approaches?

4. Many verification techniques discussed rely on properties expressible as regular languages. How can automata-based approaches be extended to verify more complex properties beyond regular languages?

5. As the size of the extracted automata increases, interpretability may decrease. Are there principled ways to control the complexity of the learned automata while maintaining fidelity?

Requested Changes
1. Clarify the novelty and positioning of the survey.
The authors should more clearly explain what differentiates this survey from previous reviews and how it contributes new insights or perspectives to the field.

2. Provide a more structured comparison of existing methods.
Adding a summary table comparing the reviewed approaches in terms of scalability, model assumptions, computational cost, and application scenarios would improve the clarity and usefulness of the survey.

3. Expand the discussion of modern architectures.
The authors may consider discussing how automata learning techniques could apply to newer sequence modeling architectures, which would broaden the relevance of the survey.

4. Strengthen the discussion of practical limitations and future directions.
A more detailed analysis of current challenges—such as scalability, benchmark design, and guarantees of correctness—would make the discussion section more informative for researchers.

---

> ### Author Response · Authors · 2026-05-27
> **Response to Reviewer hsbm**
>
> We thank the reviewer for the positive assessment and constructive feedback. The questions were an opportunity to reflect on our paper's clarity.
>
> We have addressed the noted weaknesses by: (1) clarifying the novelty and positioning in a new paragraph in Section 1, (2) adding structured comparison tables across all three application domains, (3) expanding discussion of modern architectures, and (4) strengthening our analysis of practical limitations including scalability and benchmarks.
>
> All changes are marked in purple in the revised manuscript. Minor corrections such as typos were applied without highlighting to maintain readability.
>
> ## Questions:
> 1. Many of the reviewed approaches are evaluated on relatively small RNN architectures or synthetic tasks. How well do these methods scale to modern large-scale models or real-world sequence modeling tasks?
> > This is an open research question we highlight in the discussion. For modern recurrent architectures (Mamba, RWKV), no works exist yet. However, in Luna (Song et al., 2024), the authors extract DTMCs from transformer-based LLMs (Alpaca-7b, Llama2-7b, CodeLlama-13b-Instruct). While extraction time is not reported, the number of states remains realistic, indicating potential for large-scale models. Regarding real-world tasks, many surveyed papers already apply to problems such as sentiment analysis.
>
> 2. Since automata learning produces surrogate models, how accurately do the extracted automata represent the behavior of the original neural networks in practice?
> > Accuracy is typically measured via a fidelity score: comparing the automaton's decisions against the RNN's on a dataset and averaging agreement. On real-world datasets fidelity is rarely 100%, hence automata learning currently provides underapproximations of RNN semantics.
>
> 3. The paper highlights the lack of standardized benchmarks for this research area. What characteristics should such benchmarks include to enable fair comparison of different approaches?
> > Two needs exist: (1) synthetic benchmarks with known ground-truth automata to assess algorithm correctness, and (2) real-world tasks demonstrating practical utility. Both are essential for meaningful comparisons across approaches.
>
> 4. Many verification techniques discussed rely on properties expressible as regular languages. How can automata-based approaches be extended to verify more complex properties beyond regular languages?
> > To our knowledge, this is an open research problem. Since learning algorithms lack tractable guarantees for language equivalence between automaton and RNN, the common alternative is deriving statistical guarantees from samples via Probably Approximately Correct (PAC) learning, as in Mayr et al. (2021).
>
> 5. As the size of the extracted automata increases, interpretability may decrease. Are there principled ways to control the complexity of the learned automata while maintaining fidelity?
> > Yes. Hong et al. (2022), discussed in Section 3.1, propose a passive learning method using k-means clustering initialized with a fidelity threshold during state merging, enforcing an explicit trade-off between fidelity and size.
>
> ## Requested Changes
> 1. Clarify the novelty and positioning of the survey. The authors should more clearly explain what differentiates this survey from previous reviews and how it contributes new insights or perspectives to the field.
> > Our novelty lies in three contributions now highlighted in a paragraph added at the start of "Objective and Outline" in Section 1.
>
> 2. Provide a more structured comparison of existing methods. Adding a summary table comparing the reviewed approaches in terms of scalability, model assumptions, computational cost, and application scenarios would improve the clarity and usefulness of the survey.
> > We added three tables (one per application domain) in Section 4, covering automaton type, learning style, scalability, application domains, result types, and code availability.
>
> 3. Expand the discussion of modern architectures. The authors may consider discussing how automata learning techniques could apply to newer sequence modeling architectures, which would broaden the relevance of the survey.
> > Standard recurrent architectures receive less attention with the rise of Transformers and State Space Models. We expanded the discussion of modern architectures; please see the edited paper for details.
>
> 4. Strengthen the discussion of practical limitations and future directions. A more detailed analysis of current challenges—such as scalability, benchmark design, and guarantees of correctness—would make the discussion section more informative for researchers.
> > We expanded the discussion section to cover broader research directions, notably in interpretability. We also discussed the need   for standardized benchmarks suitable for RNN, and modern architectures based on recurrence.

---

### Review · Reviewer_3H7p · 2026-04-27

**Summary Of Contributions:**

The authors provide an up-to-date review of literature from an important topic - applying automata learning to recurrent models to hopefully pave a pathway toward safe AI. Their background and early notations section is elegant, with succinct and accurate formula allowing wide accessibility to interested readers. I found the beginning pages of this survey particularly exciting and relevant to my own neural architecture research. The authors introduce three approaches to automata learning from recurrent neural networks: (1) passive, (2) active, and (3) hybrid. They then discuss applications in formal verification, model-based testing and interpretability.

**Additional Comments:**

See above in requested changes.

**Audience:**

Yes

**Audience Explanation:**

I personally found the survey to be exciting and relevant to my own neural architecture research. I plan to follow this research direction to see how it might be of use in future work, and believe this survey could be a valuable guide in navigating this field for others.

**Broader Impact Concerns:**

No ethical or broader impact concerns to report at this time.

**Claims And Evidence:**

Yes

**Claims Explanation:**

The submission is a literature review, so the authors' claims are supported by relevant peer-reviewed publications.

**Requested Changes:**

I found the organization and clarity of presentation began to degrade once I reached page 6 with the introduction of active learning. I believe this section, as well as hybrid learning, could benefit from better definitions as well as diagrams to illustrate their behavior. There are also a fair bit of acronyms throughout the paper, but I would encourage the authors to only utilize the ones that are critical to communicate their ideas from start to finish. For instance, CEGAR only appears briefly, perhaps it can not be used? Careful selection of acronyms will improve the paper's clarity. As for mathematical notation, I felt there was some inconsistency in the notation used in this survey, but I may be mistaken. For instance, the use of x (or X), capital letters to denote what appear to be vectors (e.g., page 9) when it would be more natural to assume sets or, if bold, matrices. For instance, is H_{i} the sequence up to time-step i? Is that necessary, or can the LTS encoding sequence as depicted in the same paragraph instead be: (h_{0}, y_{0}) -- x_{1} --> (h_{1}, y_{1}) ----> ... -- x_{n} --> (h_{n}, y_{n}). This format might be more intuitive, given the transitions are based on an individual element, not a sequence

I recommend the authors keep track of various alphabetic/Greek letters, their different styling, and how it relates to meaning to ensure a minimal and consistent mathematical notation is kept.

Confused by page 9 writing of the following paragraph:

"Afterwards, they define safety state properties: High confidence, Decisiveness, Robustness, and Coverage by taking the Globally modality and checking if the state predicates hold at all states of the LTS. And temporal safety properties: Long-term relationship, that checks if an RNN trained over sequences of length n can globally have comparable confidence measures on sequences of length p > n, and Memorization checks if at all states we have confidence scores close to 1 − ϵ."

More specifically, "... by taking the Globally modality and checking..." doesn't sound correct - is it meant to say "Global modality"?

The start of the next sentence, "And temporal safety properties..." seems sudden and abrupt, but then only lists "Long-term relationship". Further, the sentence continues with "... can globally have comparable confidence measures" but it may flow better with "... can have comparable global confidence measures..." (read it with either and select the best). In fact, it would be easier to understand what is going on in this paragraph by first previewing these terms, then expanding upon them:

"Afterwards, Vengertsev & Sherman, 2020 define safety state properties and temporal safety properties. The following were defined for the former: high confidence, decisiveness, robustness, and coverage. These take the global modality and check if the state predicates hold at all LTS states. (actually, you might want to better explain each of these properties than is done here) Examples of temporal safety properties are long-term relationship and memorization. More specifically, long-term relationship checks if an RNN trained over sequences of length n can have comparable global confidence measures on sequences of length p > n. Memorization then checks if at all states we have confidence scores close to 1 - ϵ."

There appears to be a lack of cohesive mathematical notation. For instance, page 10 uses lowercase s to denote a state which I am assuming relates to a finite automaton, but earlier on page 3, it is lowercase q that denotes states of FAs. Perhaps I am missing something, but are the lowercase h the hidden states of that formula? I assume they are, but it currently is not well-introduced. The lack of cohesion in the mathematical content somewhat detracts from the usefulness of the survey; currently, it reads a bit disjointed.

Additionally, some articles are elaborated quite a bit (e.g., DeepStellar page 10 to 11), but fail to clearly convey that paper's ideas; as a result, the submission doesn't currently appear to represent many articles despite a lengthy references section.

Minor suggested edits:
1. Add spacing between GRU and (Cho et al., 2014) on page 3 after definition 1 but before subsection 2.2
2. Consider changing notation F for final (or accepting states) as it slightly conflicts with state transition function (1) for RNN (if we ignore the theta)
3. There is an unnecessary comma after the em dash on page 6 just before "and closed". Actually, this sentence is unclear: "... -, and closed - The prefix of a word that is either accepted or rejected by ..." it is a bit confusing how it is written
4. End of page 7, the footnote does not have a period to mark the end of sentence.
5. Top of page 8, "Summarizing tables are given the appendix 6." should be written as "Summary tables are provided in Appendix 6."
6. Figure 5 caption, rewrite to "The Property Directed Verification approach, where filled and dotted lines mean the formula is evaluated to true and false, respectively." The reference to filled lines as straight lines implies there is no curvature, but they clearly bend.
7. Page 10: "A summarizing table is given in the appendix section ." needs to be finished.
8. A comma appears just before "See example in Figure 6.", when a period is needed.
9. End of page 12, "The discussed works are encapsulated in the appendix table ." needs to be finished.
10. Start of page 13, change "(the Tower of Hanoi)" to "(e.g., the Tower of Hanoi)" if it is just one of many reasoning problems tested on. Otherwise, consider rewriting to remove parentheses.
11. Start of page 13, "And at that stage, we know that the agent strategy is represented by that automaton." <-- avoid ever starting a sentence with "And"
12. A bit informal end to section 4 with "Check the appendix for a summary table with code links to the discussed works.", suggested rewrite: "Please refer to the appendix to review a summary table; this table also includes links to code repositories of several of the works discussed within."
13. Page 22, appendix A would have been better if the tables were not in landscape mode - it makes it more troublesome to review. If code is not available, maybe state that instead of omitting. Make the "Yes" clickable, but it is also good to have the URLs written out as you have done here.
14. Throughout the work, when referenced authors are used in the sentence, as in the above paragraph with Vengertsev & Sherman, you should remove the parentheses from the reference so the citation is inline.

---

> ### Author Response · Authors · 2026-05-27
> **Response to Reviewer 3H7p**
>
> We are deeply grateful to the reviewer for the exceptional attention devoted to our work. We sincerely appreciate it. We also want to express how encouraging it was to read that you found the survey exciting and relevant to your own neural architecture research; knowing that this work may serve as a guide for researchers like yourself is precisely the outcome we hoped for.
>
> We have carefully addressed every point you raised. All changes are marked in purple in the revised version of our work, with the exception of minor typographical corrections, which were applied without highlighting to preserve readability.
>
> ## Requested Changes
>
> 1. Clarity of active and hybrid learning sections:
> > We have rewritten the section on active learning and added an example along with a new figure to better illustrate the behavior of both active and hybrid learning (as hybrid learning depends on active and passive learning principles). We hope the revised presentation now matches the clarity you appreciated in the earlier pages.
>
> 2. Acronym usage:
> > We have removed unnecessary acronyms, including CEGAR, ANS, SUL, and RIL, retaining only those that are used consistently and critically throughout the paper. Thank you for this suggestion.
>
> 3. Mathematical notation consistency:
> > We sincerely thank you for your attentive reading and effort in identifying these inconsistencies. You were not mistaken: the notation was indeed inconsistent in several places. We have unified the notation throughout the paper, ensuring symbols are introduced when needed and used coherently. We also added two notation tables in Appendix A (referenced at the beginning of Section 2) so that readers can easily track all alphabetic and Greek letters, their styling, and their meaning.
>
> 4. Paragraph on Vengertsev & Sherman (2020):
> > We rewrote the paragraph following your structure, and we also expanded the individual definitions to give the reader a better understanding of what each property captures.
>
> 5. DeepStellar and article elaboration:
> > Your observation is well taken. We had extended the presentation of certain papers (including DeepStellar) as a means to introduce shared concepts that recur across multiple works in the section. However, we recognize that this intention was not made sufficiently explicit, and that some paragraphs related to the learning phase deviated from the section's focus. We have rewritten Section 4.2 to make these connections more explicit, and omitted content that detracted from the clarity of each paper's actual contribution.
>
> 6. Minor suggested edits (1–14):
> > We have addressed all fourteen points. Specifically: spacing corrected (1), accepting states notation changed to $Q_f$ and $Q_i$ (2), sentence restructured for clarity (3), footnote punctuation added (4), phrasing corrected (5), figure caption rewritten with "filled" instead of "straight" (6), incomplete references to appendix tables now completed (7, 9), comma replaced with period (8), parentheses adjusted with "e.g." (10), sentence no longer begins with "And" (11), informal phrasing replaced with the suggested rewrite (12), appendix tables reformatted (13), and inline citations corrected throughout (14). We are grateful for these careful observations collectively, they have significantly helped us improve the quality and readability of our work.

---

### Review · Reviewer_tfMo · 2026-05-14

**Summary Of Contributions:**

This paper surveys methods for extracting interpretable surrogate automata from RNNs using automata learning and applying them to verification, testing, and interpretability tasks. Experimental results across prior works suggest these methods are effective for robustness analysis, coverage-guided testing, and fault localization, but they still suffer from scalability limitations and weak formal guarantees.

**Audience:**

Yes

**Audience Explanation:**

The paper addresses the intersection of automata learning, recurrent neural networks, verification, testing, and interpretability, which are relevant topics for researchers working on trustworthy machine learning, formal methods, and sequential models. The survey provides a structured synthesis of passive, active, and hybrid automata learning approaches, while also discussing practical applications and open challenges such as scalability and formal guarantees.

**Broader Impact Concerns:**

I do not see major ethical concerns beyond the standard considerations associated with interpretability and verification research. In fact, the work is largely safety-oriented, as it aims to improve the reliability, robustness, and transparency of recurrent neural networks.

**Claims And Evidence:**

Yes

**Claims Explanation:**

The claims are generally supported by clear evidence, as the submission is primarily a survey paper and its main claims are about synthesizing existing work rather than introducing new empirical results. The paper supports these claims by organizing prior methods into passive, active, and hybrid automata learning, then connecting them to applications in verification, testing, and interpretability, with concrete examples from the literature.

**Requested Changes:**

1. Add a more systematic comparison table across surveyed methods, including dimensions such as automaton type, learning style (passive/active/hybrid), scalability, guarantees, and application domain. This would make the survey easier to navigate and strengthen the paper’s value as a reference.
2. Clarify the limitations of current approaches more explicitly, especially regarding scalability to modern large-scale sequence models and the lack of strong equivalence guarantees between learned automata and the original RNNs.
3. Include a short discussion comparing automata-based interpretability with alternative interpretability paradigms for sequential models, such as attribution-based or probing-based methods.

---

> ### Author Response · Authors · 2026-05-27
> **Response to Reviewer tfMo**
>
> We thank the reviewer for the careful reading and positive assessment of our paper. We appreciate the recognition that this survey addresses a number of relevant topics for researchers working on trustworthy AI. We also thank the reviewer for the feedback and suggested changes, which helped us improve the quality of our work.
>
> All changes made in response to your review are marked in purple in the revised manuscript. However, minor corrections such as typos and small phrasing adjustments were applied without purple highlighting to maintain readability.
>
> 1. Add a more systematic comparison table across surveyed methods, including dimensions such as automaton type, learning style (passive/active/hybrid), scalability, guarantees, and application domain. This would make the survey easier to navigate and strengthen the paper’s value as a reference.
>
> > We have added three tables (one for each application domain: verification, model-based testing, and interpretability) in Section 4. These tables include the automaton/learned model type, learning style, scalability, and application domain. We also included the type of results produced (e.g., which kinds of properties can be verified in the case of verification methods) and whether the code is publicly available, in which case we provide links to open-source repositories.
>
> 2. Clarify the limitations of current approaches more explicitly, especially regarding scalability to modern large-scale sequence models and the lack of strong equivalence guarantees between learned automata and the original RNNs.
>
> >  We took advantage of our summary tables presented at the end of each application subsection to provide insights on the scalability of the studied works. Since scalability is closely tied to the learning algorithms and benchmarks used, we address this in our discussion section by insisting on the need for more standardized benchmarks. Specifically, we argue that such benchmarks should go beyond simple regular languages and first target more complex synthetic languages (e.g., languages defined over large alphabets that are considerably more complex than Tomita grammars, while remaining easy to represent with automata and thus easier to visualize for debugging purposes), then extend to real-world data. We believe this is a necessary step to demonstrate the practical relevance of automata learning and whether it scales effectively. We also discuss these limitations in the modern architectures section, where we briefly cite works that have successfully applied automata learning to large models (e.g., Transformers) and compare them with modern RNN architectures such as SSMs, which we believe are more relevant for future research on this topic.
>
> 3. Include a short discussion comparing automata-based interpretability with alternative interpretability paradigms for sequential models, such as attribution-based or probing-based methods.
>
> > We thank the reviewer for this suggestion, which adds relevance to our discussion section. Comparing automata-based interpretability methods against standard approaches is indeed necessary to motivate their use. We have expanded our discussion on interpretability by first comparing automata-based methods with attribution-based and probing-based methods, highlighting the advantages and disadvantages of each. We also included more recent works showing that automata-based explanations can be easier to compute compared to other explanation types such as counterfactual explanations, and additive Shapley values. We believe our revised discussion now clearly highlights what automata-based methods can offer for the interpretability of sequential models.

---

### Decision · Action_Editor_SG3f · 2026-06-23

**Recommendation:** Accept with minor revision

**Additional Comments:**

The authors should upload a camera ready version with all changes as promised during rebuttals.

**Audience:**

Yes

**Audience Explanation:**

The topic is sufficiently interesting for those who want to better understand RNNs, which would surely be a large segment of the TMLR readership.

**Claims And Evidence:**

Yes

**Claims Explanation:**

This is a survey paper looking at the application of automata theory to RNNs. Following the reviews and replies, the paper now provides a clear and thorough review of the subject. The reviewers were unanimous on this, and all recommended acceptance.